# Virtual Joint Motion Simulator Accurately Predicts Effects of Femoral Component Malalignment during TKA

**DOI:** 10.3390/bioengineering10050503

**Published:** 2023-04-22

**Authors:** Liam Montgomery, Ryan Willing, Brent Lanting

**Affiliations:** 1School of Biomedical Engineering, University of Western Ontario, London, ON N6A 3K7, Canada; rwilling@uwo.ca (R.W.);; 2Department of Mechanical and Materials Engineering, University of Western Ontario, London, ON N6A 3K7, Canada; 3London Health Sciences Centre, London, ON N6A 5W9, Canada

**Keywords:** total knee arthroplasty, surgical malrotation, knee modelling, ligaments

## Abstract

Component alignment accuracy during total knee arthroplasty (TKA) has been improving through the adoption of image-based navigation and robotic surgical systems. The biomechanical implications of resulting component alignment error, however, should be better characterized to better understand how sensitive surgical outcomes are to alignment error. Thus, means for analyzing the relationships between alignment, joint kinematics, and ligament mechanics for candidate prosthesis component design are necessary. We used a digital twin of a commercially available joint motion simulator to evaluate the effects of femoral component rotational alignment. As anticipated, the model showed that an externally rotated femoral component results in a knee which is more varus in flexion, with lower medial collateral ligament tension compared to a TKA knee with a neutrally aligned femoral implant. With the simulation yielding logical results for this relatively simple test scenario, we can have more confidence in the accuracy of its predictions for more complicated scenarios.

## 1. Introduction

Total knee arthroplasty (TKA) is a common end-stage treatment for knee osteoarthritis performed nearly 800 thousand times annually in the US, with rates expected to increase by 143% by 2050 compared to 2012 [1]. Though generally successful at providing the patient with a pain-free knee, patient satisfaction rates following TKA remain lower than those for total hip arthroplasty (THA) [2]. Bourne et al. reported TKA satisfaction rates of only 81% compared to 89% for THA, while Mahomed et al. noted that TKA satisfaction was only 89% compared to 97% for THA [3,4]. It is not well understood why a larger percentage of unsatisfied TKA patients exist. Abdelnasser et al. found a positive correlation between component malalignment and poor patient-reported outcomes in the two years following a primary TKA [5]. Additionally, Naili et al. suggest that higher patient satisfaction may be linked to improved biomechanics after noting that patients reporting a good outcome had greater peak flexion angles and greater flexion–extension range, compared to patients reporting a poor outcome, despite similar levels of pain [6]. With component malrotation and biomechanics both being linked to patient satisfaction, there exists a need to identify the relationship between malrotation and biomechanics following TKA.

With nearly all the knee’s degrees of freedom unrestricted by bony anatomy, a key player in knee biomechanics are the ligaments. Each ligament imposes a motion restraint upon the knee joint to only allow the knee to flex and extend. For the purposes of this study, we will briefly review the posterior cruciate ligament (PCL), the medial collateral ligament (MCL), and the lateral collateral ligament (LCL). The collateral ligaments (LCL and MCL) work together to prevent medial-lateral translation, internal–external rotation, and varus–valgus rotation [7]. The PCL prevents posterior motion of the tibia with respect to the femur, as well as some external rotation of the tibia with respect to the femur, and is also responsible for initiating femoral rollback [8].

In the mechanically aligned TKA, the femoral component will be rotated externally by 3.0 ± 1.2° with respect to the femur’s posterior condyle axis (PCA); a target which has been shown to reduce the need for lateral retinacular release and decreases the risk of poor patellofemoral outcomes [9]. Siston et al. reported malrotation errors ranging from 13° internal rotation to 16° external rotation, which highlights the large opportunity for rotational malalignment in the operating room [10,11]. To emphasize the importance of femoral rotation, Sternheim et al. found that patients experienced similar relief following a revision due to malrotation compared to revision due to aseptic loosening [12].

Patient-reported outcomes and retrospective studies are important tools in improving patient satisfaction; however, knowing the biomechanical effects of malrotation could help us better understand the mechanism behind these poor patient outcomes. Joint motion simulators can be useful tools for studying the effects of malalignment, but such apparatuses can be expensive and their experiments time-consuming; thus, computational approaches are appealing. In 2011, Thompson et al. used a computational model to examine the biomechanical effects of extremely large malrotations (>10°) and found that a 15° malrotation resulted in significantly higher ligament and quadricep forces [11]. Though computational models have been used to examine the biomechanical effects of large malrotations, a gap exists in the smaller degrees of malrotation described by Boya et al. [7]. Using a simulation, in the form of a digital twin of a commercially available joint motion simulator, allows component malrotation to be simulated at a lower cost, while model results can be directly reproduced on an identical physical simulator if need be. Therefore, with an ideal femoral component external rotation being cited as 3.0 ± 1.2°, our objective is to employ a digital twin of a joint motion simulator to assess the kinematics and ligament forces of a TKA knee with femoral components over- and under-rotated by 1.5°.

## 2. Materials and Methods

### 2.1. Virtual Simulator

This study used a model developed within virtual simulation software (VIVO Sim Visualization Software AMTI, Watertown, MA, USA)—herein referred to as the virtual simulator. The virtual simulator is a digital twin of a mechanical, commercially available servo-hydraulic, 6-degrees of freedom (6-DOF) joint motion simulator (VIVO, AMTI), where each DOF can be operated with force- or displacement-control (Figure 1). Considering the individual components, the femur is capable of flexion–extension and adduction–abduction motions only; all translations and rotation about its long axis are constrained. The tibia is capable of translations in three directions, and rotation about its long axis, whereas flexion–extension and adduction–abduction are constrained. This virtual simulator can incorporate the contributions of virtual ligaments which behave as 1D non-linear elastic point-to-point springs. These virtual multifiber ligament models calculate and incorporate forces that are collinear with the ligament fibers, which are resolved to equivalent forces and moments acting across the knee [13]. This simulator and its virtual ligament system have been successfully employed in previous studies to examine knee laxity and investigate knee implant effectiveness [14,15,16]. Ligament wrapping can be enabled, which determines if a point along the length of a virtual ligament is penetrating through a wrapping surface (bone or prosthesis component). If penetration exists at any point, a single optimized point on the ligament will be extended beyond the wrapping surface. Thus, wrapping is mimicked by the software establishing a new ligament attachment site on the wrapping site that eliminates the possibility of ligament penetration through bone or component geometries. Each ligament requires femoral and tibial insertions, stiffnesses, and zero-force lengths to be defined. The zero-force length, or slack length, is defined as the length at which the ligament first becomes taught, i.e., the length at which the ligament strain is exactly zero [17]. 

### 2.2. Knee Model

The computational knee model used for this study was adapted from one developed by Guess et al. which included bony geometries and ligament parameters for a healthy right knee [18,19]. The bony anatomy of the healthy knee had to be modified to include mechanically aligned, single-radius cruciate-retaining (CR) TKA components. Component geometries were for a Triathlon CR implant (Stryker Corporation, Kalamazoo, MI, USA) which includes a femoral component, a tibial tray, and an ultra-high molecular weight polyethylene (UHMWPE) insert. All components behaved as rigid bodies with contact between the femoral component and UHMWPE insert defined with a coefficient of friction of µ = 0.04 [20,21]. Contact is determined by the virtual simulator by evaluating whether penetration exists between two surfaces (e.g., femoral component and UHMWPE insert) in the inferior–superior (IS) direction. Femoral component rotation was established by comparing the PCAs of both the native femur and the femoral condyles of the prosthesis (Figure 2). These axes were determined as the lines that connected the posterior-most points on each of the native femur’s/femoral component’s condyles. The femoral component was then rotated externally by 3° with respect to the femur’s PCA [22]. We also designed two more models with identical bony and implant geometry whose femoral components were rotated +/−1.5° for our external (+1.5°) and internal (−1.5°) model. Component placement and sizing were verified by an experienced orthopedic surgeon. 

### 2.3. Ligament Model

The ligament model used in this study was also adapted from Guess et al.’s model which contained insertions, stiffnesses, and zero-force lengths for fourteen bundles that represent seven distinct ligaments, amongst which were the PCL, superficial MCL (sMCL), and LCL. Our TKA model did not include the anterior cruciate ligament (ACL), the anterolateral ligament (ALL), the posterior oblique ligament (POL), or the deep MCL (dMCL), as these ligaments are routinely released as a part of TKA surgery [23,24]. Thus, from the Guess model, we retained the three-bundle sMCL, the three-bundle LCL, and the two-bundle PCL. This ligament model was further modified to increase the number of bundles used to represent each ligament. To create the final model that would eventually be used for biomechanical testing, two additional collateral ligament bundles and one additional PCL bundle were added to the adopted model. The added bundles had insertions midway between those of the two adjacent bundles, as well as reference strains assumed to be the average of those of the two adjacent bundles.

The given zero-force lengths had to be adapted to reference strains to be input to the virtual simulator. The reference strains (*ε_r_*) refer to the strain experienced by the virtual ligaments at the reference pose (knee in extension). Reference strains were calculated from the zero-force lengths (*l_0_*) provided by Guess et al. and the ligament lengths at the reference pose (*l_r_*) using Equation (1).
(1)$$\varepsilon_{r}=\frac{l_{r}-l_{0}}{l_{0}}\times 100\%$$

Stiffnesses used in the final model were calculated such that force and force distributions for each ligament remained similar with the knee in a distracted position that engaged all ligament bundles, regardless of the number of bundles used to represent the ligament. This study simulated the wrapping of all sMCL bundles around the proximal medial aspect of the tibia. Thus, our final model contained a wrapped, five-bundle sMCL, a five-bundle LCL, and a three-bundle PCL. A detailed explanation of stiffness calculations can be found in Appendix A, while ligament parameters used for the final model can be found in Table 1 and graphically depicted in Figure 3.

Additionally, ligament wrapping was enabled, which determines if a point along the length of a virtual ligament is penetrating through a wrapping surface (bone or prosthesis component). If penetration exists at any point, the ligament will be pushed out to lie on the wrapping surface, mimicking wrapping. This study simulated the wrapping of all sMCL bundles around the proximal medial aspect of the tibia. Thus, our final model contained a wrapped, five-bundle sMCL, a five-bundle LCL, and a three-bundle PCL.

### 2.4. Loading

Using the virtual simulator, the knee models were first guided through neutral flexion (flexion prescribed, all remaining DOFs unconstrained), and resulting knee kinematics were recorded. Laxity tests were then simulated at 0°, 15°, 30°, 60°, and 90° of flexion. The posterior laxity limits were measured by applying a 100 N posterior-directed force to the tibia, varus–valgus (VV) laxity limits were measured by applying a ±8 Nm torque, and internal/external (IE) laxity limits were measured by applying ±4 Nm torques, respectively (Markholf 2008, Weirer 2020). The orientations of the various prescribed loads are shown in Figure 4. All remaining degrees of freedom were unconstrained (0 N or 0 Nm), except for the prescribed flexion angle. These tests were repeated for every combination for each condition of malrotation: internal, baseline, and external. Laxity during all motions was calculated as the absolute difference between kinematics during neutral flexion, and kinematics during the laxity test. Additionally, each condition was put through a neutral flexion loading where no external loads were applied to the joint except for the prescribed flexion angle. For clarity, our methodology has been distilled into a flow chart (Figure 5).

### 2.5. Data Analysis

All kinematics data collected in this experiment were converted to a Grood–Suntay coordinate system and collected with respect to a reference pose (knee fully extended). The net joint contact compressive force due to ligaments, as well as the portion of this net force acting through the medial versus lateral condyles, were calculated to compare how they change for different femoral rotations. Descriptive statistics were used to compare joint kinematics and ligament tensions collected during neutral flexion and laxity tests for all femoral rotations (internal, baseline, external). 

## 3. Results

### 3.1. Compressive Ligament Forces during Neutral Flexion

Figure 6 shows net compressive forces due to ligaments acting on the knee through neutral flexion. Although the externally rotated femoral component condition initially experienced the highest compressive ligament forces, net compression was reduced relative to other alignments as the knee was flexed beyond 0°.

The change in compression experienced by both the medial and lateral compartments of the knee for the internal and external conditions when compared to the baseline condition was also examined. This force imbalance favored the medial compartment for all malrotation conditions and at all flexion angles. Figure 7 shows that compression of the medial compartment due to ligament forces was reduced in the externally rotated model compared to the baseline as the knee was flexed, whereas this force was increased in the internally rotated model. Figure 8 displays the same data for the lateral compartment. There was not a clear pattern in the changes in compressive ligament forces acting on the lateral compartment.

### 3.2. Posterior Laxity

Posterior laxity was similar across all three malrotation scenarios, as shown in Figure 9. Specifically, AP position during neutral flexion remains within 1 mm for all rotation conditions and all angles of flexion. Similarly, posterior laxity remained within 1 mm for all conditions and at all flexion angles. The greatest posterior translation of the tibia with respect to the femur for all levels of malrotation occurred at 30° of flexion, while full extension and 90° of flexion saw the tibia closest to its position during neutral flexion.

### 3.3. Varus–Valgus Laxity

The more externally rotated femoral implants lead to a more varus knee during both laxity tests and neutral flexion, particularly during late flexion (Figure 10). Notably, beyond 15° of flexion, the more externally rotated components resulted in a more varus the knee, with this difference becoming more pronounced as the knee became more flexed. However, due to a similar shift during both tests, the relative varus and valgus laxities remained within 0.9° between all three malrotation conditions throughout flexion. 

### 3.4. Internal–External Laxity

IE kinematics during neutral flexion and IE laxity tests are described in Figure 11 for all three conditions of model rotation. Overall internal laxity was greater than external laxity, regardless of IE position, and similar for all three rotation conditions during neutral flexion. During late flexion with an internal torque applied to the knee, the internally rotated component leads to a reduction in laxity.

## 4. Discussion

The objective of this study was to assess the effects of femoral component malrotation on knee joint kinematics and ligament tensions following TKA by using a virtual joint motion simulator. We predicted that component malrotation would affect IE mechanics during extension and affect VV kinematics in late flexion as the rotation of the femoral component in extension will manifest as VV rotation once the knee is flexed to the point of the condyles being in contact with the UHMWPE insert.

Femoral component rotation manifested a clear effect on net ligament compression as the knee was flexed, whereupon the more externally rotated components resulted in a lower compressive force. When an externally rotated component is flexed, the sMCL insertions will be closer together, leading to a reduced strain and thus lower tension in extension. 

With no other forces acting on the knee, the medial side will be tighter in both the healthy and the TKA knee due to medial side ligaments exerting higher forces than their lateral counterparts [19,20]. For this reason, the change in compressive ligament forces due to component malrotation had a more noticeable effect on the medial compartment of the knee.

During VV laxity tests, as well as neutral flexion, a greater external rotation of the femoral component resulted in a more varus knee position in extension. Inversely, a more internally rotated femoral component results in a more valgus knee position in deep flexion. In a more varus knee position, the sMCL insertions will be closer together, leading to a reduced strain and thus lower tension in extension. In the native knee, the sMCL will exert a compressive force on the joint but it will also exert a varus torque, and similarly, the LCL exerts a valgus torque. Therefore, the varus knee positional in flexion caused by a more externally rotated femoral component will increase the LCL strain and decrease the sMCL strain.

Interestingly, component malrotation seemed to have little influence on knee kinematics during both neutral flexion and valgus laxity tests. This is likely due to the dominance of the force exerted by the sMCL during these motions. During valgus laxity tests, the medial side of the joint is opened increasing the strain on the medial ligaments and reducing the strain on the lateral side ligaments. Further, Jeffcote et al. demonstrated that the MCL strain will be greater than that of the LCL during neutral flexion in a balanced TKA knee [25]. Further, the greater stiffness of sMCL bundles compared to the stiffnesses of the LCL bundles proves that the sMCL will be exerting a much higher force except for motions where the LCL is forced to be engaged such as varus laxity. Our results show that during motions where the sMCL is clearly more engaged than the LCL, the effect of rotational malalignment is difficult to discern. This suggests femoral component rotations between 1.5° and 4.5° will have a noticeable kinematic effect during motions where the LCL exerts a greater strain than the sMCL.

We also showed that femoral component malrotation had an effect during internal rotation laxity testing, with effects varying at different flexion angles. However, a pattern is noticeable at points of late flexion beyond 60°, with a more externally rotated femoral component resulting in a more internally rotated laxity limit. There is an opposite but much smaller effect during external rotation, and no effect at all during neutral flexion. The kinematic effect of malrotation during IE laxity tests is easily explained by the externally rotated model’s ligaments exerting a weaker external tension to resist the applied internal torque.

Overall, component malrotation did not have a noticeable effect on AP kinematics during either neutral flexion or posterior laxity. However, a poorly aligned femoral component in the sagittal plane could lead to a change in kinematics within the AP degree of freedom [26].

A limitation of this study was that only one model was tested. Though this was a parametric study, demonstrating similar trends in other subjects could strengthen our results. Additionally, the virtual simulator is limited in the motions and loads that can be automated, meaning data collection for clinically relevant flexion angles greater than 90° would need to be collected manually and at a great time cost. Furthermore, the loads we used were not indicative of loads experienced by the knee joint during weight-bearing activities [27]; thus, future work should examine ligament forces and engagement patterns during weight-bearing activities of daily living, such as gait and stair climbing.

## 5. Conclusions

This study used a computational TKA knee model within a virtual joint motion simulator to assess the effect of small errors in femoral component rotation. Little investigation had previously been undertaken on the effects of these smaller errors; however, the results aligned with our understanding of knee biomechanics and the relationship between the knee’s VV rotation, IE rotation, and flexion angle. The virtual simulator has not been employed in many previous studies and thus the simple experimental design allowed us to confirm its pre-clinical usability.

In summary, the virtual simulation employed in this study yielded results that are generally intuitive and agree with previous studies. This builds confidence for using the model for more complicated scenarios where the response of the knee may be harder to intuit.

## Figures and Tables

**Figure 1 bioengineering-10-00503-f001:**
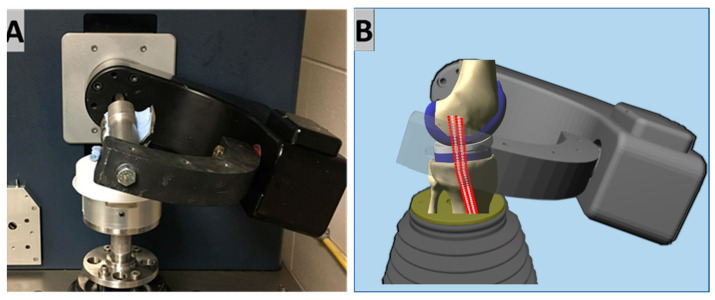
A comparison between the physical joint motion simulator (AMTI VIVO, (**A**)) and the virtual joint motion simulator (VIVO Sim Visualization Software, (**B**)). While fixtures look different, the shape and position of the actual prosthesis components related to the mechanical axes of the simulators can be identical between the two platforms.

**Figure 2 bioengineering-10-00503-f002:**
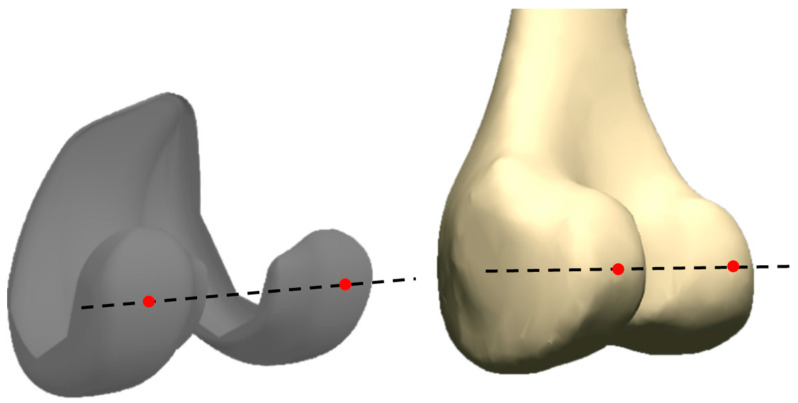
Posterior condyle axes of the femoral implant (**left**) and the femur (**right**) for a right knee model.

**Figure 3 bioengineering-10-00503-f003:**
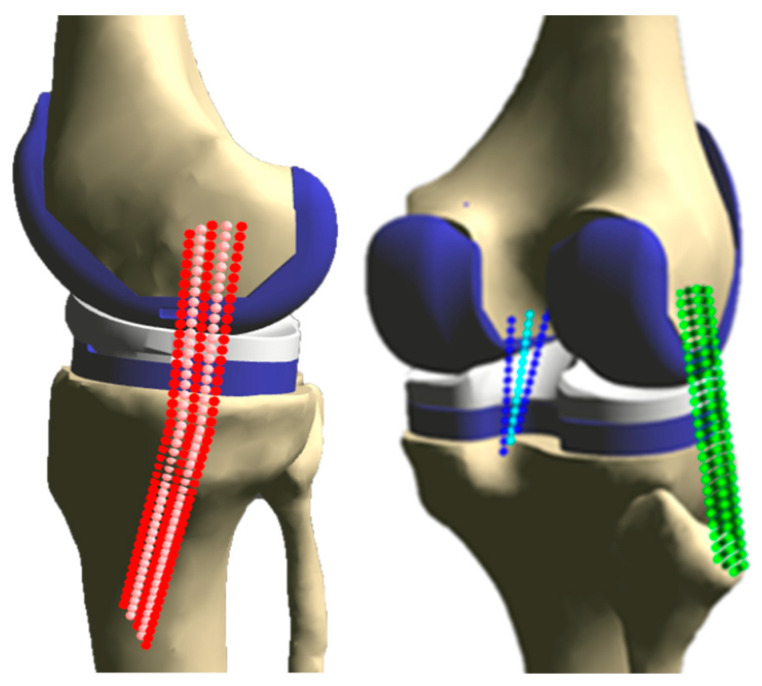
Medial (**left**) and posterolateral (**right**) views of the virtual ligament model of the right knee model used in this study. The sMCL (red) and LCL (green) are represented with five bundles each, and the PCL (blue) is represented with three bundles.

**Figure 4 bioengineering-10-00503-f004:**
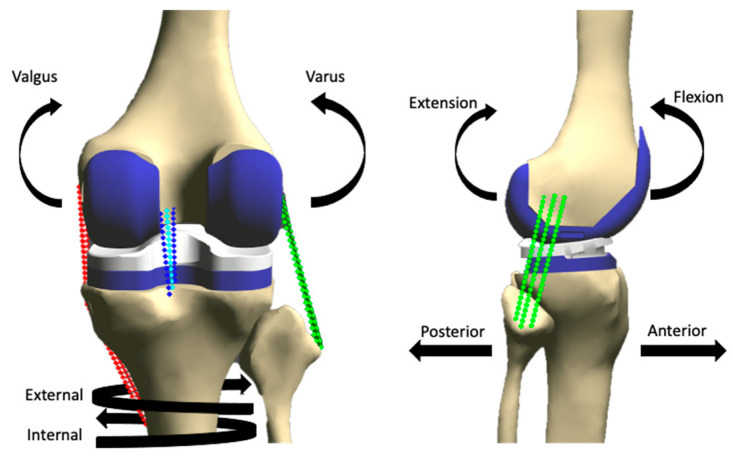
Orientations of loads applied to a right computational knee model with various conditions of femoral component malrotation.

**Figure 5 bioengineering-10-00503-f005:**
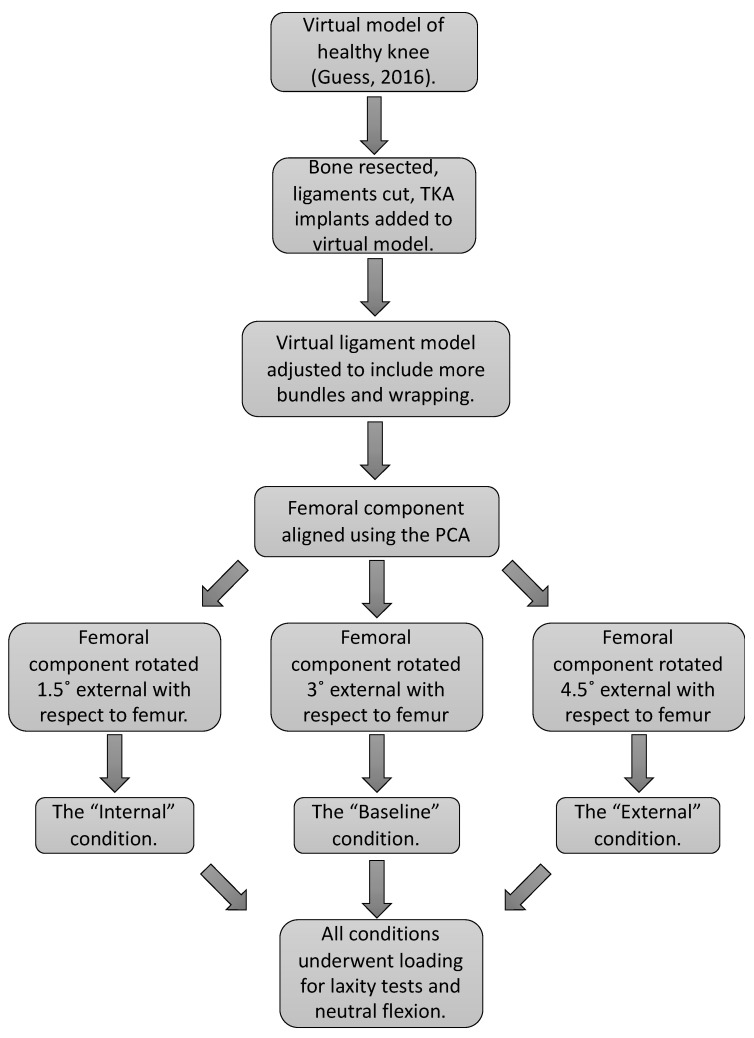
A summary of steps taken to adapt a previously published knee model into a virtual knee model used to investigate the biomechanical effects of femoral component malrotation.

**Figure 6 bioengineering-10-00503-f006:**
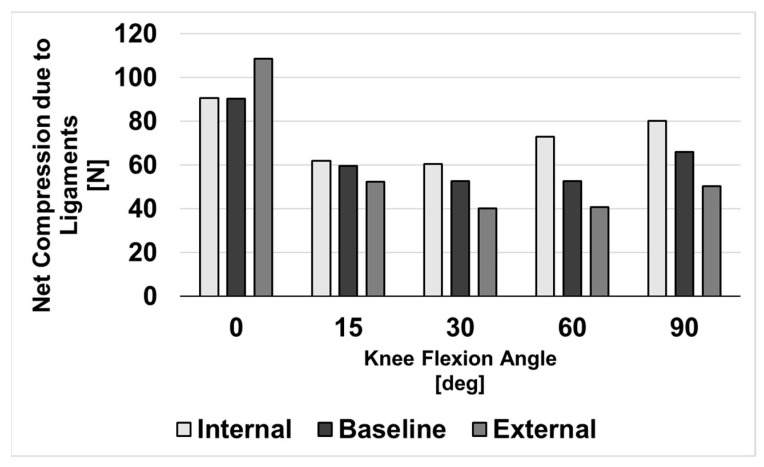
Net compression due to ligaments for three conditions of femoral component malrotation.

**Figure 7 bioengineering-10-00503-f007:**
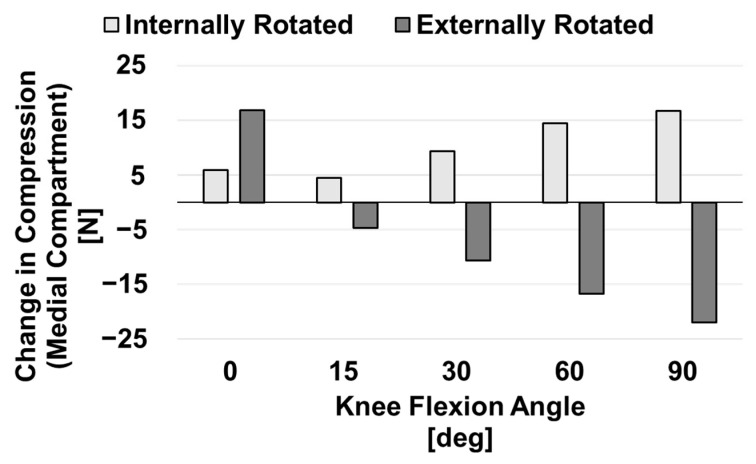
Difference in compressive ligament forces experienced by the medial compartment of malrotated models when compared to those of the correctly aligned baseline model.

**Figure 8 bioengineering-10-00503-f008:**
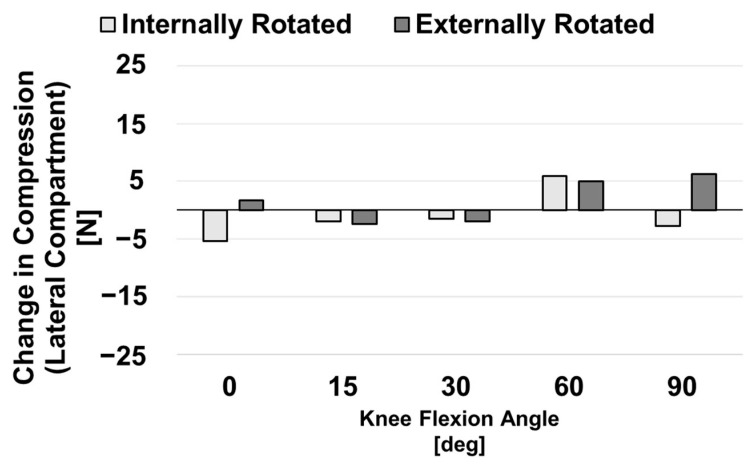
Difference in compressive ligament forces experienced by the lateral compartment of malrotated models when compared to those of the correctly aligned baseline model.

**Figure 9 bioengineering-10-00503-f009:**
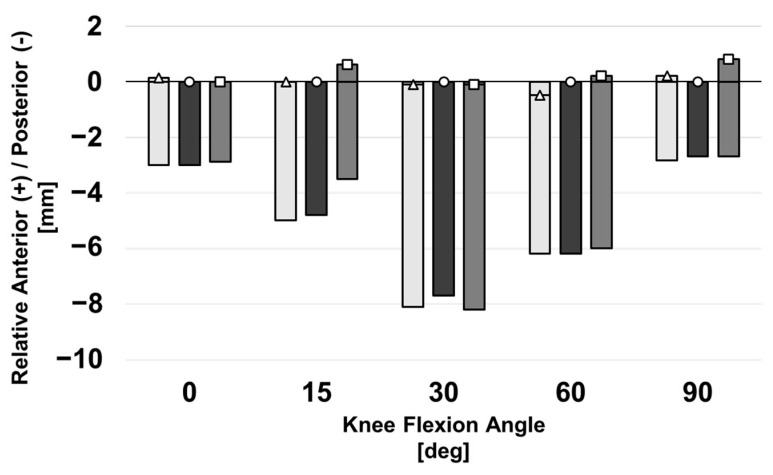
AP kinematics for three computational knee models with varied femoral implant rotation during posterior laxity tests. AP position during neutral flexion is denoted by a triangle (internal), circle (baseline), or square (external). Shaded bars depict laxity; the posterior displacement occurring in response to a posterior-directed load. All positions are shown relative to the baseline model’s kinematics during neutral flexion. Negative values denote a tibial position that is posterior with respect to its reference pose.

**Figure 10 bioengineering-10-00503-f010:**
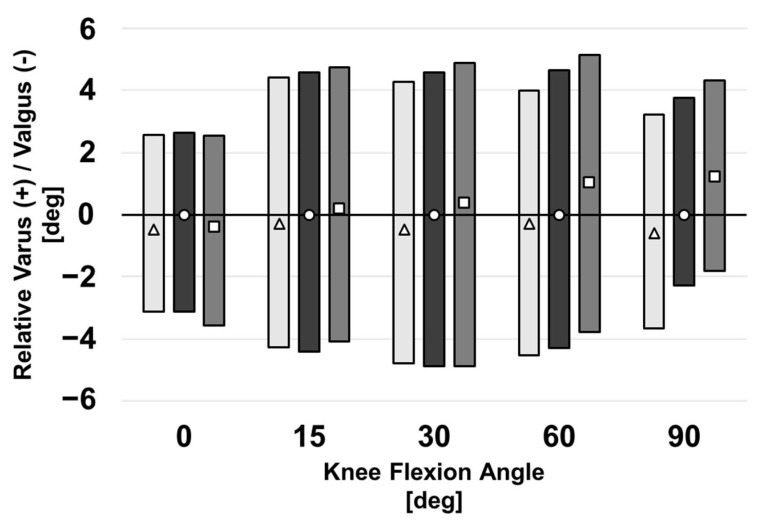
VV kinematics for a computational knee model with varied femoral implant rotation during varus and valgus laxity tests. VV position during neutral flexion is denoted by a triangle (internal), circle (baseline), or square (external). Shaded regions depict the varus and valgus laxities about the neutral positions. All positions are shown relative to the baseline model’s kinematics during neutral flexion. Negative values denote a valgus VV position.

**Figure 11 bioengineering-10-00503-f011:**
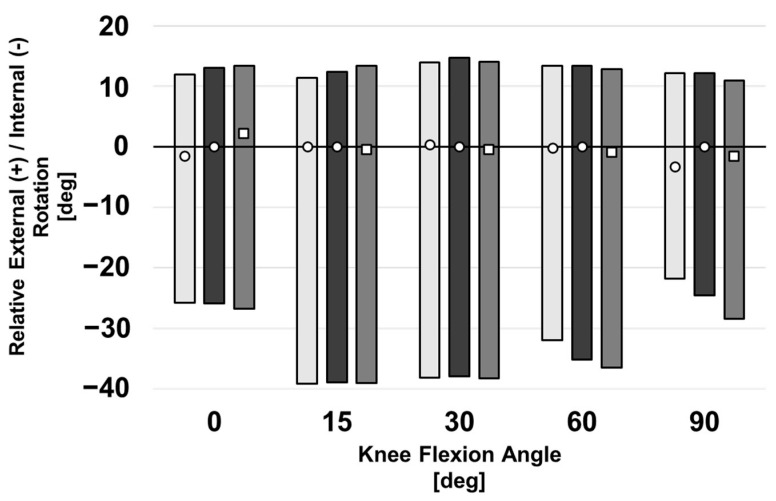
IE kinematics for a computational knee model with varied femoral implant rotation during internal and external laxity tests. IE position during neutral flexion is denoted by a triangle (internal), circle (baseline), or square (external). Shaded regions depict the internal and external rotatory laxities about the neutral positions. All positions are shown relative to the baseline model’s kinematics during neutral flexion. Negative values denote an internally rotated tibia with respect to the reference pose.

**Table 1 bioengineering-10-00503-t001:** Ligament parameters used in this study.

Ligament ^1^	Stiffness (N/ɛ)	Reference Strain (%)
aLCL	1157	−2.66
amLCL	1171	0.68
mLCL	1175	4.02
mpLCL	1172	2.66
pLCL	1182	1.29
a-sMCL	1469	−4.30
am-sMCL	1603	0.10
m-sMCL	1481	4.50
mp-sMCL	1509	4.47
p-sMCL	1105	4.44
aPCL	7841	−28.6
pPCL	1026	−26.3

^1^ Ligament prefixes for the LCL, sMCL, and PCL: a = anterior, am = anterior—middle, m = middle, mp = middle-posterior, p = posterior.

## Data Availability

The data presented in this study are available on request from the corresponding author. The data are not publicly available due to privacy concerns.

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
