# Peer review of "Virtual Joint Motion Simulator Accurately Predicts Effects of Femoral Component Malalignment during TKA"

_bioengineering, 2023, doi:10.3390/bioengineering10050503_

Round 1

Reviewer 1 Report

The article presented is interesting. The study aims to employ a digital twin of a joint motion simulator to assess the kinematics and ligament forces of a total knee arthroplasty (TKA) knee with femoral components.

There are many areas of improvement for this paper, which are listed below: 

1.       It should be included a flow chart of the methodology. This will help visualize in an easier way all the necessary steps to reproduce your simulation.

2.       In table 1, it is indicated ligaments’ stiffness. However, there is an issue with that values. It is reported that the stiffness is a force divided into a strain. Therefore, the stiffness unit must be a ratio between a force and a deformation.

3.       How is validated the numerical model?

4.       The software allows adding a contact condition? If the answer is no. How is the software considered the interaction between the material that grips the component and the bone?

5.       Because of the arguments exposed in point 2, you should check the results and discussions to corroborate all the reported information.

6.       It should be added a conclusion section.

7.       In appendix A. It must number all the referenced equations.

8.       It should be added an equation describing all the stiffness parameters. In my opinion, the ligaments´ stiffness probably behaves like a tension element, depending on the elasticity, area, and length modulus.

Reviewer 2 Report

Thank you for submitting an interesting manuscript.

I suggest several things to revise as follows.

- In all 3D model figures, it should be stated that the 3D model of femur is right side. Orientations should be annotated with arrows and directions (e.g., anterior, lateral, inferior).

- In the 3D model figures, it would be better to indicate the directions of varus and valgus with 3D arrows.

- All the "mid" should be replaced with "middle."

- The footnote of Table 1 is confusing. For example, "am – anterior—mid" should be changed to "am = anterior and middle" or something else. "amsMCL" should be changed to "am-sMCL" or something else.

- More literatures about the clinical and anatomical researches should be cited.

Reviewer 3 Report

Manuscript ID: bioengineering-2313758

Title: Virtual Joint Motion Simulator Accurately Predicts Effects of Femoral Component Malalignment during TKA

Review comments:

This study uses a digital twin of a commercially available joint motion simulator to evaluate the effects of femoral component rotational alignment.

The topic of the paper is interesting, and the work has scientific merit and clinical relevance, I am attaching a few comments, with the view to enhance the paper.

1. The validation of the physical joint motion simulator (AMTI VIVO) and the virtual joint motion simulator should be presented.

2.  How obtains the zero-force length of the ligaments?

3.  How define the baseline condition?

Round 2

Reviewer 2 Report

Thank you for the sincere revision.